# Increasing atmospheric dryness reduces boreal forest tree growth

Ariane Mirabel [1,2,3] ✉, Martin P. Girardin [2] ✉, Juha Metsaranta[4], Danielle Way [1,5,6,7] & Peter B. Reich [8,9,10]

Rising atmospheric vapour pressure deficit (VPD) associated with climate change affects boreal forest growth via stomatal closure and soil dryness. However, the relationship between VPD and forest growth depends on the climatic context. Here we assess Canadian boreal forest responses to VPD changes from 1951-2018 using a well-replicated tree-growth increment network with approximately 5,000 species-site combinations. Of the 3,559 successful growth models, we observed a relationship between growth and concurrent summer VPD in one-third of the species-site combinations, and between growth and prior summer VPD in almost half of those combinations. The relationship between previous year VPD and current year growth was almost exclusively negative, while current year VPD also tended to reduce growth. Tree species, age, annual temperature, and soil moisture primarily determined tree VPD responses. Younger trees and species like white spruce and Douglas fir exhibited higher VPD sensitivity, as did areas with high annual temperature and low soil moisture. Since 1951, summer VPD increases in Canada have paralleled tree growth decreases, particularly in spruce species. Accelerating atmospheric dryness in the decades ahead will impair carbon storage and societal-economic services.

Climate change will alter boreal forests capacity to store carbon, all the more so because they are dominated by cold-tolerant species and are warming faster than most other land areas (0.2 °C to 0.5 °C per decade over 1961–2015)[1,2]. Warming is expected to alter the net annual carbon uptake of boreal forests through changes to temperature-related variables (growing season length, drought severity, vapour pressure deficit, freeze frequency) that affect tree mortality, recruitment, physiology and growth[3–5]. Across Canada's boreal forest, land carbon storage capacity varies due to local and regional differences in climate, vegetation, soils, surficial geology, and disturbance regime[6]. The complexity of the boreal environment and the diversity of boreal species produce notable contrasts in productivity responses to warming[6–8]. Across boreal Canada, forest productivity is anticipated to decrease in the west because of water stress, and increase northeastward where temperature is currently the main factor limiting tree growth[9–11]. Such mechanisms have been raised as potential drivers of forest greening or browning in Canada[12,13]. Warming can thus enhance plant growth in cool and wet environments, but impair it above a threshold of water stress[7,14]. Additionally, differential trajectories among tree genera, species and ecoclines are possible, depending on their stress tolerance and plasticity[15–18]. Changes in growth rates of boreal forests are of great concern due to the critical roles these

[1]Department of Biology, University of Western Ontario, London, Ontario, Canada. [2]Natural Resources Canada, Canadian Forest Service, Laurentian Forestry Centre, Quebec City, QC, Canada. [3]UMR DECOD (Ecosystem Dynamics and Sustainability), Institut Agro, IFREMER, INRAE, Rennes, France. [4]Natural Resources Canada, Canadian Forest Service, Northern Forestry Centre, Edmonton, AB, Canada. [5]Research School of Biology, The Australian National University, Acton, ACT 2601, Australia. [6]Environmental & Climate Sciences Department, Brookhaven National Laboratory, Upton, New York, USA. [7]Nicholas School of the Environment, Duke University, Durham, NC, USA. [8]Department of Forest Resources, University of Minnesota, St. Paul, MN 55108, USA. [9]Hawkesbury Institute for the Environment, Western Sydney University, Penrith, NSW 2753, Australia. [10]Institute for Global Change Biology, and School for the Environment and Sustainability, University of Michigan, Ann Arbor, MI 48109, USA. ✉e-mail: Ariane.mirabel@gmail.com; martin.girardin@canada.ca

forests play in global carbon storage, human population support and economic-industrial services[19].

Atmospheric vapour pressure deficit (VPD), the difference between the amount of water vapour in the atmosphere and the potential amount held at saturation[20], is a major temperature-related determinant of plant physiology. Without a corresponding increase in the actual amount of atmospheric water vapour, VPD will increase with warming because the saturation vapour pressure is a curvilinear function of air temperature[21]. High VPD can induce high leaf and soil water loss, increasing the risk of plant water deficits and stress[22,23]. Water deficits increase xylem tension and tissue cavitation: these negative effects are minimized by stomatal closure, which decreases water loss at the cost of carbon uptake, leading to growth reductions and eventually tree mortality[20,21,24,25]. From a hydrometeorological perspective, higher atmospheric dryness also increases atmospheric demand for water from the land surface, increasing evaporation and reducing available soil moisture[26]. Hence, plant growth responses to atmospheric dryness can result from direct effects on stomatal conductance and indirect effects on soil moisture through increasing evapotranspiration[27].

Tree species have evolved different strategies to cope with atmospheric dryness and therefore have differential responses to VPD changes, modulated by environmental conditions[20]. There is thus a need for in-situ, multi-species comparisons to quantify these differences across a broad range of environmental conditions[25,28]. Recent remote sensing studies have produced mixed results about the relative role of VPD on productivity and carbon uptake[26,29]. Tree-ring studies offer an important opportunity to understand VPD impacts, especially pertaining to aboveground carbon uptake. To date, tree-ring analyses directly linking changes in dryness with boreal forest growth have only focused on a few species and areas[30]. Integrating tree-ring sampling into national forest inventories has enabled large-scale ecological assessments related to forest health and carbon cycling, including in Canada[31–33]. Quantifying large-scale tree growth responses to VPD would improve our understanding of forest growth responses under climate change and reduce uncertainties in the model predictions used to evaluate strategies for mitigating and adapting to climate change.

Here, we assess boreal forest responses to changes in atmospheric VPD using a well-replicated tree-ring network covering Canada's forests over the period 1951–2018. The growth-VPD relationships enabled mapping of spatially-explicit VPD responses across Canada's boreal zone. We then explored the main drivers of differential growth responses to VPD, including species, local precipitation and temperature, elevation, and tree age and size. Finally, we determined how VPD and growth are changing over time for the most responsive species. Our results indicate that increased VPD has already had a broad-scale negative effect on tree growth in Canada's forests over recent decades, specifically for younger trees and widespread spruce species.

## Results

### Tree response to atmospheric vapour pressure deficit

Utilizing tree-ring data obtained from 32,189 trees spanning Canada (Fig. 1), we developed 4931 Generalized Additive Mixed Models (GAMMs) based on species and site factors. These models were employed to evaluate the impact of VPD on the annual basal area increment (BAI, the increment in cross-sectional area of trees). We achieved convergence in 3559 (72%) of the species-site GAMM models, indicating successful outcomes. The average goodness-of-fit between year-to-year observed growth fluctuations and growth predicted from site-specific growth-VPD models was $r^2 = 0.51$ ($\sigma \pm 0.21$, $n = 3559$ convergent models). Among the convergent species-site GAMMs, 58% (2057 models) showed a significant relationship between BAI and $VPD_t$, $VPD_{t-1}$, or both across the nine species analyzed (*Picea mariana*, *Picea glauca*, *Pinus banksiana*, *Populus tremuloides*, *Pinus contorta*,

*Pseudotsuga menziesii*, *Picea engelmannii*, *Abies lasiocarpa*, and *Pinus resinosa*).

Among the convergent species-site GAMM models, 31% (1,096 models) had a $t$-value for the relationship between annual BAI and VPD of the year of ring formation ($VPD_t$) achieving the $p < 0.05$ threshold. Of these 1,096 significant relationships, 752 (about two-thirds) were negative, indicating that increasing $VPD_t$ is detrimental to tree growth for these sites-species combination (Fig. 2, Table S1). The majority of negative $t$-values were found near warm, dry margins of the boreal forest in southcentral Canada (i.e, southern part of the Boreal Shield ecoregion), specifically for *Picea glauca* and *Picea mariana* (Figs. 1–3, and Supplementary Materials Fig. S1). A third of the significant BAI-$VPD_t$ $t$-value were positive (344 models), indicating that increasing $VPD_t$ was positively related with current year tree growth (Fig. 2, Table S1). The positive $t$-values were found in western Canada, i.e Boreal Cordillera, and in cooler moister areas of central and eastern Canada, i.e Hudson Plain and Taiga Shield (Figs. 1–3, Fig. S3). Both negative and positive $t$-values were found for all species except *Pseudotsuga menziesii*, the distribution area of this species does not overlap the previously cited ecoregions. Furthermore, a significant relationship with VPD of the year prior to ring formation ($VPD_{t-1}$) was observed in 47% of the convergent species-site GAMM models (1,660 models). In these models, $t$-values were almost exclusively negative (96%, Fig. 2, Table S1). Collectively, our findings suggest that VPD, whether from the previous year or the current year, had a significant and detrimental impact on BAI in approximately 51% of the 3,559 species-site combinations where models converged (roughly 37% of the initial 4,931 candidate species-site models). Based on these observations, VPD responsiveness of tree growth was found in one-third to half of the species-site combinations. The consistency of the results was maintained in the partial BAI-VPD GAMM models, which aimed at controlling for the impact of summer soil moisture index (SMI) on growth, as demonstrated in the Supplementary Materials.

### Identify the determinants of VPD growth response

We assessed the determinants of growth response to VPD through the relationship between the significant $t$-values of growth-VPD relationship and seven environmental, biological and forest structure variables, namely site elevation, mean annual temperature (MAT), mean annual precipitation (MAP), summer SMI, tree species, mean tree age and BA at the site level.

Among the significant variables, average depth in the random forest decision trees was lowest for species identity. Species identity was selected as root node in 158 of 500 trees, meaning it is the first variable determining the direction and strength of growth-VPD $t$-values (Fig. 4, Table 1). The species showing more negative VPD growth responses were, in order of most to least sensitive, *Pinus resinosa*, *Pseudotsuga menziesii*, *Pinus contorta*, *Picea glauca*, *Picea mariana*, and *Pinus banksiana*, while *Populus tremuloides*, *Picea engelmanii* and *Abies lasiocarpa* showed a less negative response to VPD. Site-level mean tree age displayed the second lowest average depth and was selected as a root node in 68 of 500 decision trees (Fig. 4, Table 1). Mean tree age was generally positively correlated with $t$-values (Fig. 5), meaning a more negative response to VPD occurred in young to mature trees (0–100 years) compared to mature and old trees (100–250 years); a low sample size hinders the capacity to infer the typical growth response to VPD beyond 250 years of mean tree age. Site-level mean BA displayed high average depth and very low MSE, meaning it makes a minor contribution to $t$-value variations.

Local mean annual temperature (MAT) was the primary environmental and climatic determinant of $t$-values, with the third lowest average depth among random forest trees and second highest selection as root node (for 117 of 500 trees). MAT was negatively correlated with $t$-values, meaning VPD had a more negative effect on growth in higher MAT environments or during warmer years (Fig. 5). Long-term

**Fig. 1 | Description of tree-ring dataset. a** Distribution of sample sites. The background colour on the map at left illustrates the distribution of the aboveground biomass (AGB) across Canada's forests[S1], the distribution of sample sites for the tree-ring dataset encompassing the nine sampled species is represented by points. Canadian Ecoregions are illustrated on the map at right, with the hemi-boreal zone overlaid. Yearly temporal distributions of sampled (**b**) trees, (c) sites, and (**d**) species proportions.

mean summer SMI showed the fourth lowest average depth and was chosen as the root node in 96 decision trees (Fig. 4, Table 1). Mean summer SMI was positively correlated with *t*-values, meaning the most detrimental effects of VPD were seen in sites with low available soil water (Fig. 5).

**Spatiotemporal changes in VPD and growth of widespread tree species**

Central and northern Canadian boreal regions have experienced significant increases in summer VPD since 1951 (Fig. 6a). The affected regions extend across the ranges of *Picea mariana* and *Picea glauca* (Fig. 1, Fig. S1). At these locations, mean VPD values increased from slightly above 0.5 kPa in the 1950s–1960s to >0.6 kPa in the 2000s. We found that percent growth changes (GC, see Methods) in terms of BAI, averaged by species, were linearly and inversely related to prior-summer VPD (Fig. 6b, c). From the slope of this relationship, we estimated that for every ~0.1 kPa increase in VPD, there was approximately a 10 to 11% decrease in mean BAI throughout the geographic ranges of these spruce species (Fig. 6b, c). Decreases in tree growth from 1951 to

the present were apparent in both species, paralleling the increases of summer VPD (also see Fig. S5). Although the northern part of the region was sparsely sampled among our ~3500 sites, because the ring-width of both species is negatively affected by high prior-summer VPD values (Figs. 3, 6), we might also expect a negative growth response for these species in this region in the decades ahead. We also examined the same relationship for two other species that showed significant growth sensitivity to VPD, *Pinus contorta* and *Picea engelmannii* (Fig. 6d, e). Again, the percent change in growth averaged across *Pinus contorta* was linearly correlated with prior-summer VPD, but to a much lesser extent than in the two spruce species. For *Picea engelmannii*, there was no evidence for a negative relation of range-wide averaged growth change to prior-summer VPD (*r* = –0.18 with 95% CI [–0.44; 0.12]), but rather a moderate positive relationship to current-summer VPD (Fig. 6d).

## Discussion

We show that increasing atmospheric drought has negatively impacted growth in all nine major boreal forest species assessed. The main

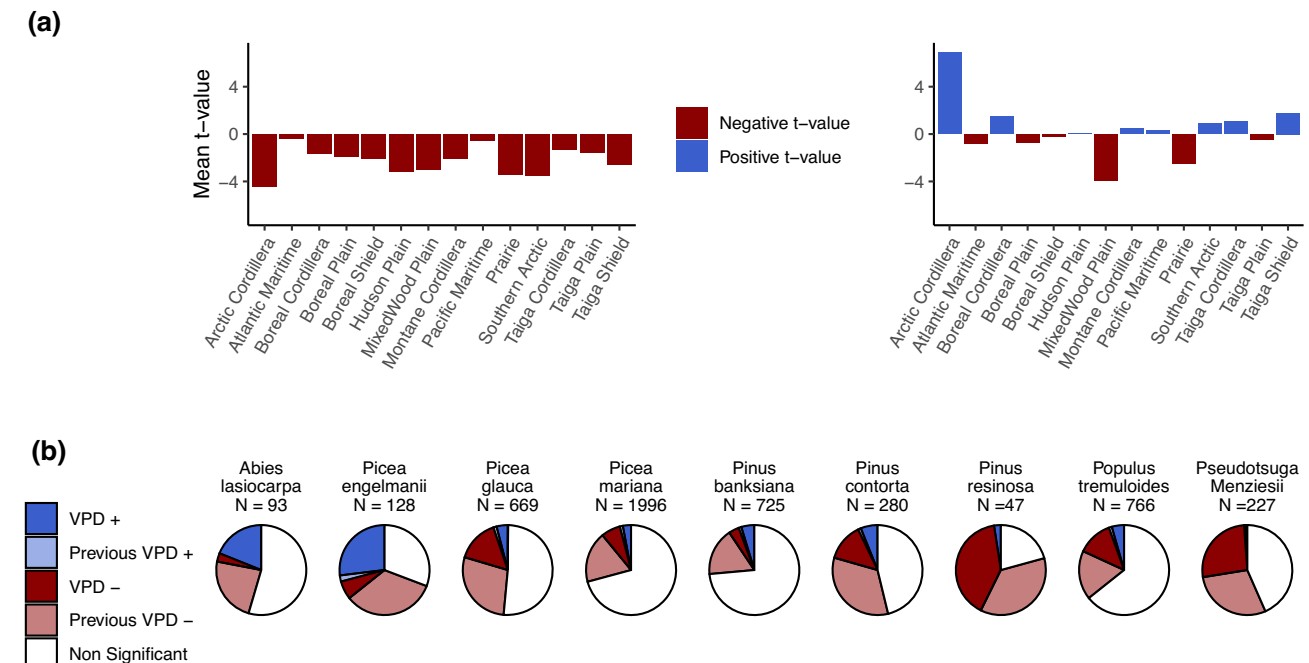

**Fig. 2 | The relationship between annual growth fluctuations estimated from tree rings (BAI) and vapour pressure deficit (VPD). a** Means of *t*-values by ecoregions for $VPD_{t-1}$ (left) and $VPD_t$ (right); and (**b**) proportions of non-significant, positive and negative *t*-values for $VPD_{t-1}$ and $VPD_t$, among convergent models, with the number N of corresponding sites. The density distribution of all *t*-values is illustrated in Fig. S2.

response to increasing VPD was a first growth reduction for the current growing season in some species, followed by a second, larger, growth reduction the following year. This observed pattern was consistent across Canada and particularly pronounced in younger trees. These findings corroborate those observed in several temperate forests, specifically the lag between carbon uptake, measured from flux towers, and carbon allocation to wood the following year, assessed by tree-ring measurements. This lag, while not well understood, is often assumed to be controlled by allocation to non-structural carbon (NSC) storage, which is induced during years of favourable growth conditions and drawn upon during years of drought[34,35]. Similar lag effects have also been seen with regard to tree growth responses to decreases in SMI, which have been attributed to the influence of deep soil moisture memory[36].

To optimize the balance between carbon gain and water loss, trees attempt to balance between water uptake and losses regulated by leaf water potential and associated stomatal conductance[37,38]. Increases in VPD and subsequent drought risk typically induce a decrease in stomatal aperture and conductance to regulate leaf-level transpiration. Stomatal sensitivity to VPD is typically higher in *Abies lasiocarpa*, *Pinus contorta*, *Populus tremuloides*, and *Pseudotsuga menziesii*, which maintain more constant leaf water potential via stomatal closure (more isohydric), and lower in *Picea mariana* and *Picea engelmannii* which allow a greater drop in leaf water potential (more anisohydric species)[21,39–43]. That being said, a clear pattern in tree growth response to VPD matching this iso/anisohydric continuum was not observed, with isohydric species being either among the most sensitive (*Pseudotsuga menziesii* or *Pinus contorta*) or the less sensitive to VPD (*Populus tremuloides* and *Abies lasiocarpa*). The absence of a clear pattern could be due to species growing in different environments across this broad spatial area, making it difficult to isolate a species effect. The iso/anisohydric pattern might also be confounded with differences in species water-use strategy, linked for example to crown and rooting system architecture, as well as stem capacitance[39,44].

Variation in the strength of tree responses to VPD was not just determined by species, but also by MAT (an effect that persisted even after removing the indirect effect of soil moisture) and by local summer SMI. This growth sensitivity to VPD confirmed that warming and drying impact boreal forest growth by increasing tree sensitivity to atmospheric water demand[20,25]. Heightened growth sensitivity to VPD at higher temperatures and lower soil moisture is also consistent with a potential physiological mechanism; data for more than a dozen species show acclimation to warmer temperatures and lower soil moisture leads to a more conservative stomatal behaviour that saves more water at the cost of gaining less carbon[45].

In contrast, some sites, largely contiguous with eastern Boreal and Taiga Shields and in the Boreal Cordillera, showed a positive growth response to increasing VPD. This positive response to VPD is likely limited to sites where growth is enhanced by a temperature increase while VPD remains low without becoming stressful, typically sites with low mean annual temperatures, excess soil moisture, or short growing seasons[7,20,45]. In particular, excess water in the soil and a shallow water table can contribute to hypoxia or anoxia (i.e oxygen deficiency or absence) in belowground plant tissues, a phenomenon that ultimately leads to a decrease in root hydraulic conductance and tree growth[46]. Atmospherically enhanced evapotranspiration with warming could mitigate these negative effects and lead to a positive effect of rising VPD on tree growth. Additionally, in colder parts of the species' ranges, the warmer temperatures associated with higher VPD could alleviate cold limitations of growth more than a quite modest increase in VPD would limit stomatal conductance. Spatial differences in the VPD-growth response corroborate the heterogeneous vegetation greening and browning patterns observed in northern areas[12,13].

The strength of tree responses to VPD depended on average tree age in the sampled site. The growth response to rising VPD was most negative for young trees, and the strength of the response decreased steadily with average site age until reaching a plateau around an average tree age of 200 years. There have been reports that wood with juvenile anatomical traits may be more vulnerable to cavitation than mature wood owing to thinner tracheid walls[47,48]. The need for NSC storage after a drought event would hence be higher at this early stage, consistent with lower overall biomass and a not-yet fully developed

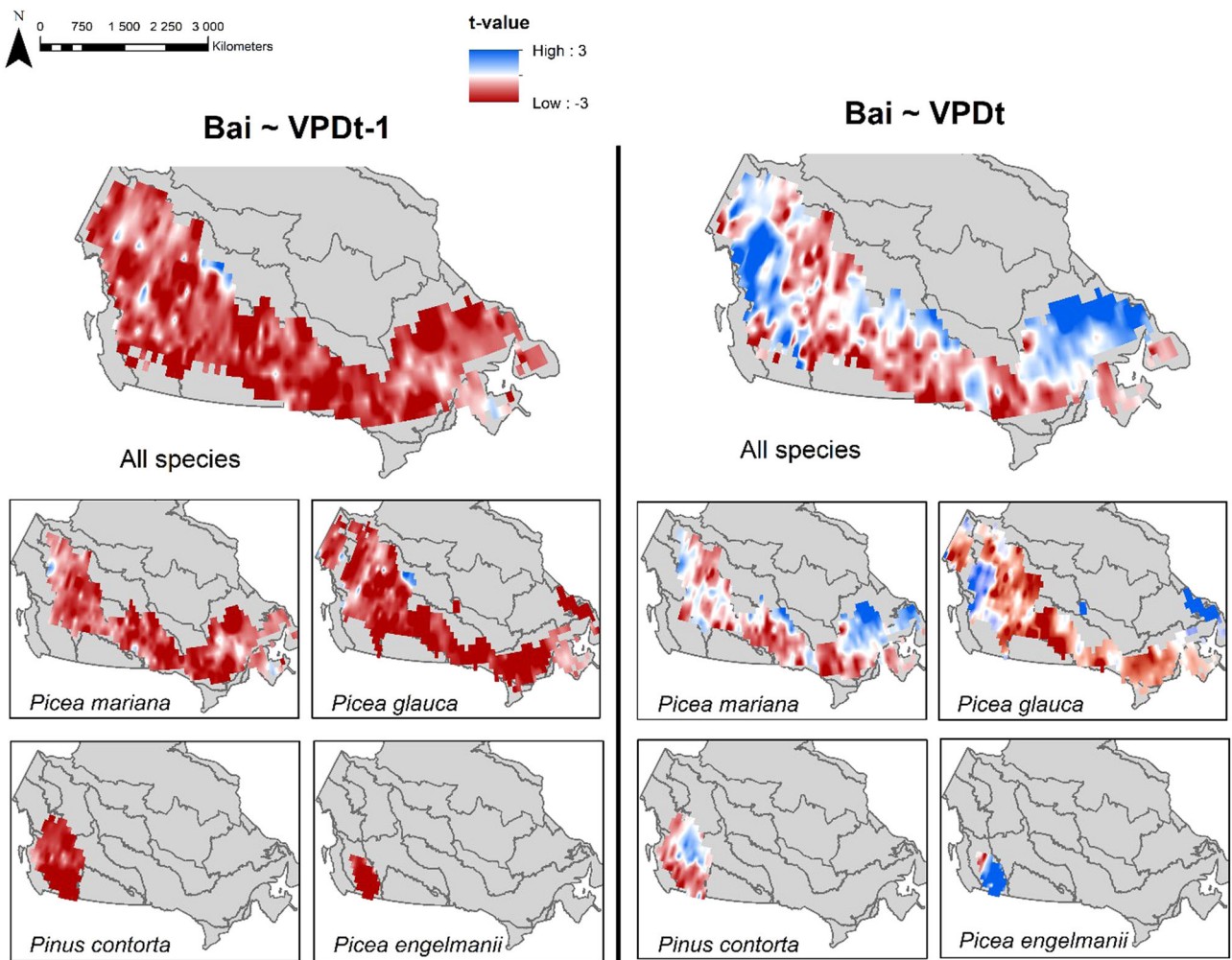

**Fig. 3 | Pointwise *t*-values of the regression between annual growth fluctuations estimated from tree rings (BAI) and vapour pressure deficit (VPD) for the species *Picea mariana, Picea glauca, Pinus contorta* and *Picea englemanii*.** Maps are displaying site-species *t*-values; a bidimensional interpolation was performed on a spatial resolution of 1 × 1 degree, using the inverse distance weighting method based on the 12 closest neighbours. Interpolations were bounded using boreal mask (all species map) and species distribution area (species maps)[80,82,83].

root system[37]. Stomatal response to VPD is known to change with tree age, along with tree growth sensitivity to drought, which decreases trees' sensitivity to water stress over time[41,49]. This varying growth response, with young trees being more susceptible to high VPD, carries significant implications for forest policies and management, particularly in the context of climate change[19,50]. An intensified influence of VPD with climate change on trees' early-life stages has the capacity to modify the success of post-disturbance tree regeneration, which plays a critical role in boreal carbon dynamics[51].

The extent of the area showing sensitivity to atmospheric drought is much larger than some modelling studies and satellite imagery analysis had suggested[15,52,53]. Such a broad extent likely reflects the multiple mechanisms linking growth to atmospheric dryness, both directly through stomatal conductance reducing growth or indirectly through lower soil moisture levels[20,21,24,26,29,54]. The relationship linking growth to atmospheric dryness noted here may also highlight the significance of temperature-induced growth stress as a potential explanation for the 'Divergence Problem' noted in high latitude and elevation forests[55]. The 'Divergence Problem' refers to a phenomenon where the relationship between tree growth and temperature decoupled or weakened from the late 20th century onwards. Regarding indirect growth effects through changing soil moisture levels, however, additional assessments of our data suggested that VPD growth

effects were not confounded by co-variation in soil moisture in the majority of studied sites. For all species, the amount of significant VPD effects on growth remained after removing the partial effect of soil moisture on growth increment, with approximately 35% of the species-site models showing a significant relationship (see Supplementary Materials and Fig. S4). These findings are consistent with recent reports suggesting that stomatal sensitivity to VPD represents the primary plant response to rising VPD[21,26].

Climate change is expected to lead to an overall increase in temperatures across Canada (https://atlasclimatique.ca/) and although an increase in spring or early summer rainfall is expected, this would not compensate for the warming-induced increase in VPD for some regions[1]. However, the specific impacts of climate change on VPD depend on a variety of factors, including changes in humidity, evaporation, and precipitation patterns[23]. Our results indicate that tree growth across Canada is strongly related to VPD, with this growth response varying with tree age, and that growth in widespread species like *Picea mariana, Picea glauca* and *Pinus banksiana* is already being suppressed by increasing atmospheric drought. We anticipate that further warming and precipitation changes over the next century will continue to hamper the growth of these common boreal species, and hence decrease forests' resilience, carbon storage capacity, provisioning of fibre and wood, and environmental services[6,19,56]. The

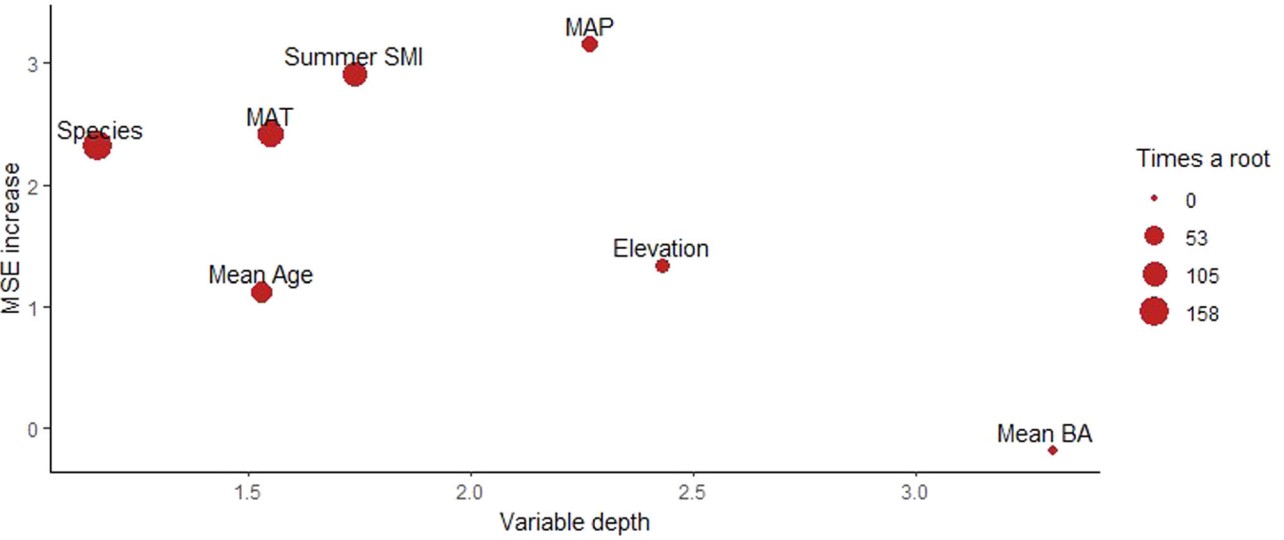

**Fig. 4 | Output of random forest algorithm predicting *t*-values from the seven environmental, tree species and forest structure variables.** The *x*-axis retrieves the variable depth in the tree, and dot size represents variables occurrence as root node (i.e more frequent root node occurrence equals larger dot); the *y*-axis retrieves variables' importance, measured as an increase in decision tree mean square error (MSE) when the variable is randomized. Results based on bootstrap with 500 decision trees (also see Table 1). The seven predictive variables are Species, Mean Age, Mean Basal Area (Mean BA), Summer Soil Moisture Index (Summer SMI), Mean Annual Temperature (MAT), Mean Annual Precipitation (MAP) and Elevation.

understanding of tree growth responses to increasing VPD should foster insights for developing adaptive measures aimed at improving forest resistance to drought conditions[42,57].

## Methods
### Study area
The study area covers eleven forested ecozones (Ecological Stratification Working Group (ESWG), 1996) in Canada (Fig. 1). These ecozones are the Montane Cordillera in the west coast, Boreal Cordillera and Taiga Cordillera that contain the Canadian Rockies, the Boreal Plains and Taiga Plains located east of the Rockies, and Prairies, Taiga Shield, Boreal Shield and Hudson Plains in the continental interior to the east. Bounded by three Great Lakes in the east are the Mixedwood Plains, and adjacent to the east coast is the Atlantic Maritime (Fig. 1). The climate in the boreal/hemi-boreal zone is predominantly high-latitude continental, with long cold winters, short cool summers, and relatively low annual precipitation, but with significant regional variation[9]. Summer mean air temperature averages from 10 °C in the Taiga Cordillera to 14 °C in the Atlantic Maritime, whereas winter mean air temperature ranges from −22.0 °C in the Taiga Cordillera to −1.5 °C in western Montane Cordillera. Annual precipitation totals average

from 200-500 mm in the Taiga Plains to over 4,000 mm in western Montane Cordillera.

### Climatic data
Daily maximum and minimum temperature (°C), precipitation (mm) and relative humidity (%) were obtained for the period from 1950 to 2018 using BioSIM v10.3 software, which interpolates site-specific estimates from all of Environment and Climate Change Canada's historical daily weather observations[58,59] (compiled weather data available at ftp://ftp.cfl.forestry.ca/regniere/Data/Weather/Daily/; last accessed 2023-06-20). Using daily minimum and maximum temperature as well as precipitation data, we employed the BioSIM software to calculate the daily vapour pressure deficit (VPD, in kPa) following the approach described in Allen et al.[60]. For dewpoint temperature calculation, BioSIM utilizes the methodology outlined in Kimball et al.[61] and leverages daily temperature and precipitation data. The soil moisture index (SMI, in % of water holding capacity) was estimated in BioSIM using the quadratic-plus-linear (QL) formulation procedure described in Régnières et al.[58], which accounts for water loss through evapotranspiration (simplified Penman−Monteith potential evapotranspiration) and gain from precipitation. Following Girardin et al.[10,31], we established the parameters for critical and maximum available soil water as 300 mm and 400 mm, respectively. A low SMI is an indication of low available soil water at the site. We acknowledge that the SMI metric used is a simplified representation of reality and entails inherent uncertainty due to the lack of detailed soil attribute representation for parameterizing site-specific water holding capacity. Daily temperature, precipitation, VPD, and SMI data were all interpolated to a 1° x 1° grid ($n = 1705$ grid points). In contrast to many other regions worldwide, Canada has a limited density of weather stations (Fig. S6), which hampers the ability to derive precise climatological information at specific sites. Although broader-resolution gridded data may not adequately capture elevational climate gradients, they do offer the advantage of homogeneity and long-term time series, which are essential for establishing climate−growth relationships[62]. VPD and SMI data were averaged for the summer season (June to August) by year, and attributed to the closest tree-ring sampling sites' grid point. In addition, long-term means of annual climatologies (1951–2018) were also computed for each site (mean annual January to December

## Table 1 | Output of the random forest algorithm

| Variable | Average minimum depth | Average MSE increase | Occurrences as root node |
|---|---|---|---|
| Elevation | 2.43 | 1.33 | 25 |
| MAP | 2.27 | 3.16 | 31 |
| MAT | 1.55 | 2.42 | 117 |
| Summer SMI | 1.74 | 2.91 | 96 |
| Species | 1.16 | 2.31 | 158 |
| Mean age | 1.53 | 1.12 | 68 |
| Mean BA | 3.31 | -0.18 | 5 |

Average depth of variables in trees, average increase in dataset mean squared error (MSE) after variable permutation, and variable occurrences as tree root node. Results based on bootstrap with 500 decision trees.
*MAP* mean annual precipitation, *MAT* mean annual temperature, *BA* basal area, *SMI* soil moisture index.

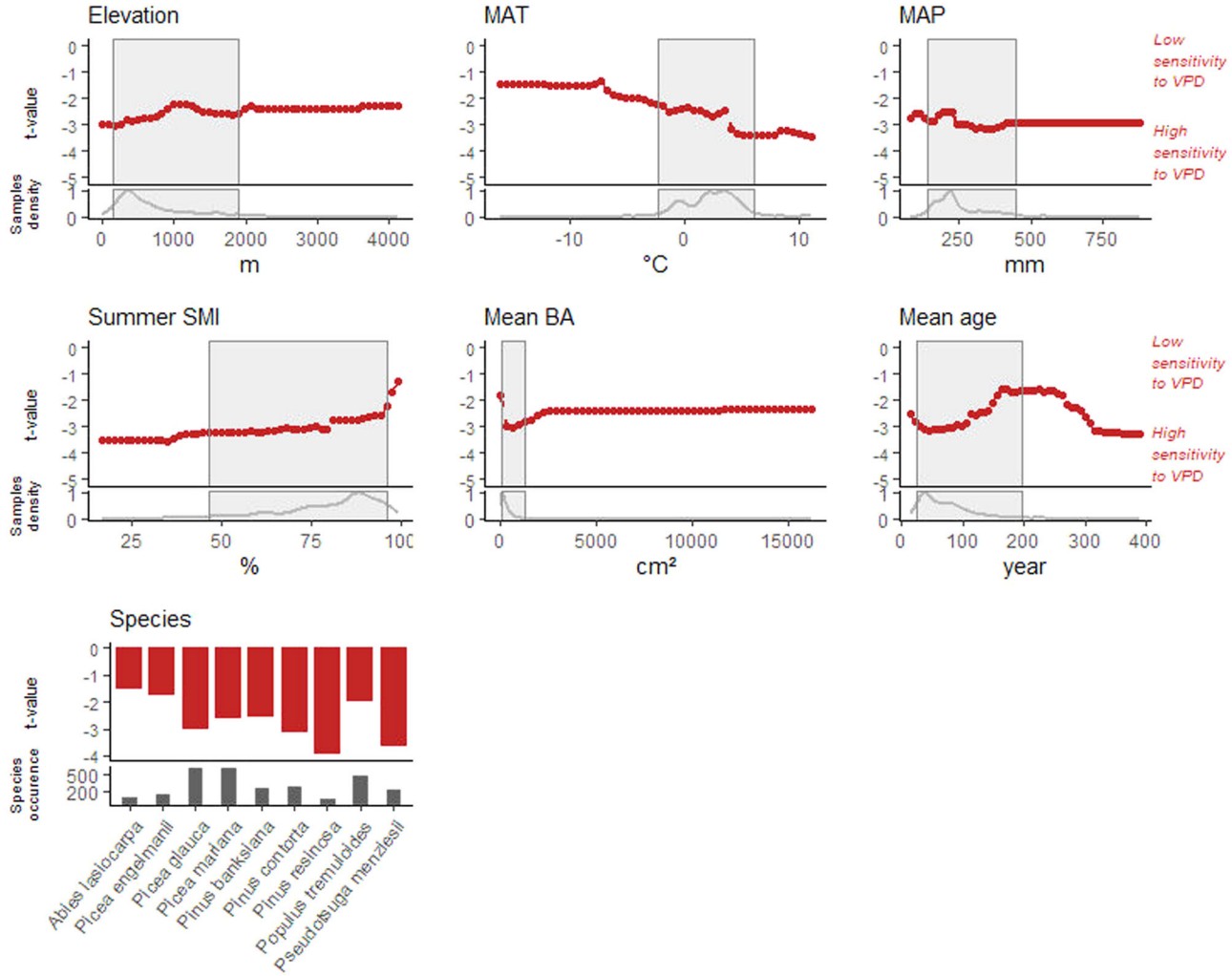

**Fig. 5 | Partial-dependence plots showing the marginal effects environmental, forest structure and tree species features have on the predicted sensitivity of growth to VPD across Canada's boreal forest.** The partial dependence functions were developed using a random forest algorithm and illustrated whether the relationships between the target and the features are linear, monotonic or more complex. The thick-red lines illustrate the predicted *t*-value changes by the random forest model, which are influenced by alterations in the features displayed on the *x*-axis. The lower curves represent sample density, while the grey-shaded areas indicate 95% sample coverages. Species is considered as a categorical feature, with the partial-dependence function illustrated by red bars; the number of sampled trees per species is illustrated by the grey bars. The seven predictive variables are Species, Mean Age, Mean Basal Area (Mean BA), Summer Soil Moisture Index (Summer SMI), Mean Annual Temperature (MAT), Mean Annual Precipitation (MAP) and Elevation.

temperature, MAT; mean annual January to December sums of precipitation, MAP; and mean summer soil moisture index, SMI).

## Tree-ring data

Annual tree-ring width data were retrieved from the Canadian Forest Service Tree-Ring repository (CFS-TRenD 1.0)[31], developed with the goal of combining data from different sources and making them available in a consistent format for large-scale analyses. CFS-TRenD contains measurements from 40,206 samples from 4,594 sites and 62 tree species. The primary national-scale dataset in CFS-TRenD 1.0 is increment cores sampled since 2001 during the establishment of Canada's National Forest Inventory (NFI)[63]. The NFI network, designed to represent species distributions and their range of growing conditions in Canada, comprises 6,010 core samples from 870 sites. Other important data are more regional and include data from a network of permanent sample plots established by the Alberta Biodiversity Monitoring Institute (ABMI), a network of temporary sample plots established by the Ministère des Ressources Naturelles et de la Faune du Québec (MFFPQ; Programme d'inventaire écoforestier nordique)[31,64], and the Climate Impacts on Productivity and Health of Aspen (CIPHA)

network[65]. Additional contributions to CFS-TRenD are from smaller-scale projects carried out by individual researchers, and data extracted from the International Tree-Ring Data Bank (ITRDB).

Analyses were limited to the nine dominant species (seven genera) in the dataset (Fig. 1) to maintain a sampling density that balances local variations in growth across sampling sites within regions, and detects growth sensitivity to climate at the regional scale. These nine species represent 80% of the samples (32,189 trees) in the CFS-TRenD repository: *Picea mariana* (black spruce; 25% of the dataset with 1.0 $e^4$ trees sampled), *Picea glauca* (white spruce; 12%, 4.7 $e^3$ trees), *Pinus banksiana* (jack pine; 11%, 4.4 $e^3$ trees), *Populus tremuloides* (trembling aspen; 10%, 3.9 $e^3$ trees), *Pinus contorta* (lodgepole pine; 7%, 2.9 $e^3$ trees), *Pseudotsuga menziesii* (Douglas fir; 7%, 2.8 $e^3$ trees), *Picea engelmannii* (Engelmann spruce; 4%, 1.7 $e^3$ trees), *Abies lasiocarpa* (subalpine fir; 2%, 9.3 $e^2$ trees), and *Pinus resinosa* (red pine; 2%, 8.7 $e^2$ trees). We excluded *Abies balsamea* (balsam fir, mostly dominant in the eastern ecozones) because its growth dynamics are strongly influenced by spruce budworm (*Choristoneura fumiferana*) defoliation[18,66]. All of the nine species occur in Canadian boreal/hemi-boreal forests[6]. Half of the sites included two or three sampled trees, and 83% of the

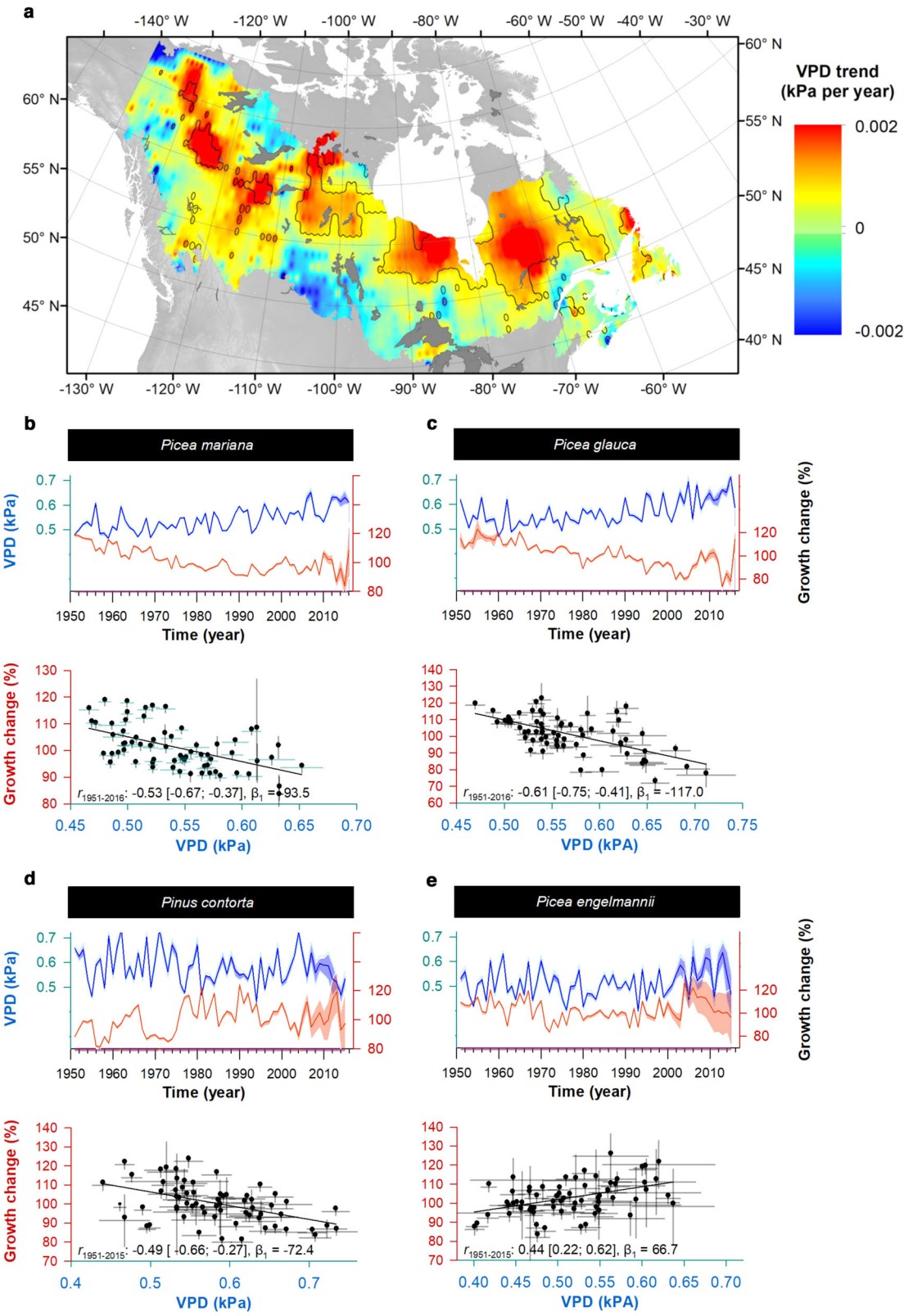

**Fig. 6 | Changes in the annual growth of *Picea mariana*, *Picea glauca*, *Pinus contorta* and *Picea englemanii* relative to summer VPD. a** Temporal trends in summer VPD (kPa / year) since 1951, measured daily temperature and precipitation using Kimball's method[61]. Percent growth change (in red) averaged across all sites relative to (**b**–**d**) prior- and (**e**) current-summer VPD (in blue). Shaded areas represent the 95% confidence interval. Sample sizes for panels (**b**–**e**) were respectively 9481, 4522, 2859, and 1632 trees. Annual growth changes are percent deviation from predicted values generated by the generalized additive mixed models representing the BAI variations unrelated to tree development stage (see Methods, Eq. 3). Correlative relationships (*r*) between growth changes and summer VPD; the 95% confidence intervals around *r* were computed from a bootstrapping technique that accounted for autocorrelation and trend in data[78], and regression slope ($\beta_1$) from ordinary-least-square regressions.

sites included a single species. Fifty percent of the trees displayed between 40 and 90 measured tree rings (i.e 25th and 75th percentiles of the age distribution). To ensure the accuracy of the measured tree rings and the assignment of the correct calendar year to each tree ring, both visual and statistical quality control procedures were implemented[10,31]. Individual contributors to CFS-TRenD typically conducted quality control using the COFECHA program[67], or analogous approaches that assess tree sample correlations against species-specific reference chronologies. Subsequently, the harmonization of the different contributions was performed within the R v4.2.2 environment[31]. In an analysis of sample coherency, Girardin et al.[31] found high spatial synchronicity in the interannual growth fluctuations among most samples.

### Responsiveness of tree growth to vapour pressure deficit
**Calculation of basal area increments.** Tree-level average ring-width series were converted to annual basal area increments (BAI; cm$^2$ yr$^{-1}$), which is recognized to be closely related to tree productivity[68]. BAI was estimated from the relation:

$$\text{BAI}_t = \pi R_t^2 - \pi R_{t-1}^2 \tag{1}$$

where $R_t$ and $R_{t-1}$ are the stem radii (cm) at the end and beginning of a given annual ring increment. Minimum tree age was determined from ring counts, starting from the outermost ring. Considering the atypical response to environmental drivers, particularly the influence of overstory tree competition on juvenile stems, tree rings formed during the initial ten years were excluded from the analysis. This decision was made based on the understanding that these early rings often exhibit distinct growth patterns that are less representative of the major environmental drivers[69].

**Growth and climate relationships.** Species-specific growth-VPD relationships were determined using generalized additive mixed models (GAMM) fitted at each site between the log-transformed BAI series, tree basal area of the year prior to ring formation and atmospheric VPD values during summer (June, July and August) of the prior and current year of ring formation, with tree considered as a random effect:

$$\log\left(\text{BAI}_{jk}^t\right) = \alpha_{jk} \cdot \log\left(\text{BA}_{jk}^{t-1}\right) + s(\text{age}^t) + \beta_{jk}^1 \cdot \text{VPD}^t + \beta_{jk}^2 \cdot \text{VPD}^{t-1} \\ + \text{corAR1}_{jk}\left(\sim t \mid \text{Tree}_{ID}\right) + \text{Tree}_{ID} + \epsilon_{jkt} \tag{2}$$

where $j$ stands for the species, $k$ for the site, and $t$ for the year. BA is the basal area, BAI is the basal area increment, *age* is the age in years, and $s$ is a cubic regression spline smoothing parameter whose degree of smoothness was determined through an iterative fitting process. Temporal autocorrelation was considered with AR1, an autoregressive term of order 1 accounting for year $t$ and $Tree_{ID}$ (each tree's unique identifier). Approximately 20% of the sampled cores either did not contain tree pith or were estimated to be located more than a centimeter away from it. Since we do not always have access to the original samples or scanned images, we were unable to apply a correction method for the pith offset. Nevertheless, we anticipate that the tree-level random effect (TreeID) would account for this missing information. If the error magnitude is important or if there is substantial noise resulting from forest management activities or abiotic/biotic disturbances, the convergence of the species-site model is anticipated to be unsuccessful. As a result, the affected site will be excluded from subsequent analyses. The significance of variables at the 5% level was determined from *t*-tests in GAMM models (*t*-value's $p < 0.05$). The growth model was fitted using the 'mgcv' v1.8.41R package[70].

**Drivers of growth sensitivity to VPD.** Environmental, tree species and forest structure variables responsible for the observed distribution in

VPD-growth relationships (i.e *t*-values) were assessed as follows. These analyses were guided by the hypothesis that the strongest negative responses to VPD would be observed at sites exposed to high temperatures or with low soil water availability. We also postulated that there is differentiation among species' responses to VPD resulting from their evolutionary strategy for stomatal conductance. We considered four environmental variables (elevation, mean annual temperature (MAT), mean annual precipitation (MAP), and summer SMI), tree species, and two forest structure variables (mean site age and basal area (BA)). First, we used the Random Forests (RF) algorithm, a machine learning method adapted to complex and potentially non-linear relationships[71]. Our RF was based on the bootstrap of 500 training decision trees aggregated to predict the *t*-values of the significant ($p < 0.05$) growth-VPD$_t$ or growth-VPD$_{t-1}$ relationships, using the seven predictor variables as the potential explanatory variables. The running and interpretation of the RF were performed using 'randomForest' v4.7.1.1 and 'randomForestExplainer' v0.10.1 R packages[72,73]. Variable importance was measured as the decrease of unscaled mean square error (MSE decrease) observed when the variable is randomized, and averaged among training trees[74]. We retrieved variables' average depth among training trees and their occurrence as the first node. Finally, to get the direction of the relationship between variables and *t*-values, we drew the partial dependence plot representing the marginal effect of each variable on the *t*-values. To untangle the direct VPD effect of atmospheric dryness on stomatal conductance from its indirect effect on soil moisture, we examined partial BAI-VPD GAMM models obtained after controlling for the effect of soil moisture index (SMI) on growth (see Supplementary Materials and Fig. S4).

**Trends in VPD and growth.** Annual fluctuations in growth over time were computed for *Picea mariana, Picea glauca, Pinus contorta and Picea engelmanii* that are among the primary harvested species in Canada, and which geographic distribution is particularly threatened by climatic changes[19]. Annual growth fluctuations were obtained from the detrending of BAI series that removed growth trends due to tree age and size[75]. We detrended time series of BAI using species-specific GAMMs fitted to the log-transformed BAI:

$$\log\left(\text{BAI}_j^t\right) = \beta_{jk} \cdot \log\left(\text{BA}_j^{t-1}\right) + s(\text{age}^t) + \text{corAR1}_j\left(\sim t \mid \text{Tree}_{ID}\right) \\ + \text{Tree}_{ID} + \epsilon_{jt} \tag{3}$$

Annual growth changes (GC), expressed as the percent deviation from predicted values generated by the GAMMs, were then computed following Girardin et al.[76]. GC was average by species and year to show yearly temporal variability in species-specific GC since 1951, see Fig. S5 for models fitting statistics. The average GC 95% confidence interval was computed at the species level, for each calendar year, by correcting the effective degrees of freedom ($n'$) based on first-order (i.e, lag = 1) autocorrelation estimates of Moran's I[77] (moran.test function in R). Correlations ($r$) between growth changes and VPD were computed from a bootstrapping technique that accounted for autocorrelation and trends in data[78], and regression slope ($\beta_1$) from ordinary-least-square regressions.

Linear trends of summer VPD were examined for 1951–2018 using least squares linear regressions. A derivation of a *t*-test with an estimate of the effective sample size accounting for serial persistence in data returned the slope's significance against the null hypothesis that the trend is zero[79].

### Reporting summary
Further information on research design is available in the Nature Portfolio Reporting Summary linked to this article.

## Data availability

The weather data generated to support the finding of this study are freely accessible through Environment and Climate Change Canada's portal (https://climate.weather.gc.ca/) and the BioSIM server (https://cfs.nrcan.gc.ca/projects/133). The tree-ring datasets have been deposited in the Natural Resources Canada TreeSource repository https://treesource.rncan.gc.ca/en. Tree-ring datasets may be available under restricted access for third-party data, access can be obtained through contact details included in the TreeSource repository. The data that support the plots within this paper and other findings of this study are deposited in the FigShare repository https://doi.org/10.6084/m9.figshare.24260554.

## Code availability

All relevant software and R-functions that support the methods of this study are referred to in the "Methods" section (see package vignettes for details). The detailed code is available in the GitHub repository https://github.com/ArianeMirabel/Dendrochronology.git, https://doi.org/10.5281/zenodo.8410445[80].

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

## Acknowledgements

The authors acknowledge the important contribution of the National Forest Inventory program, from which most of the data were obtained. Other important data contributors include the provincial governments of Alberta and Quebec and the International Tree-Ring Data Bank. This work was made possible thanks to the financial and in-kind support provided by the Canadian Forest Service of Natural Resources Canada.

## Author contributions

M.P.G., D.W., and A.M. conceived this work. A.M. and M.P.G. wrote the original manuscript. A.M., M.P.G., D.W., J.M., and P.B.R. discussed and interpreted the results and contributed to the manuscript.

## Competing interests

The authors declare no competing interests.
