## [Peer Review File · Nature Communications]

Increasing atmospheric dryness reduces boreal forest tree growthEditorial Note: This manuscript has been previously reviewed at another journal that is not operating a transparent peer review scheme. This document only contains reviewer comments and rebuttal letters for versions considered at *Nature Communications*.

REVIEWERS' COMMENTS

Reviewer #3 (Remarks to the Author):

Second Review of Mirabel et al. "Increasing atmospheric dryness reduces boreal forest tree growth"

I appreciate that the authors have made an effort to alleviate my concerns. The required additional information regarding the climate data has now been added to the Methods section and (some of) the figures have been improved.

Based on the information provided, I think it is very likely that the results are robust w.r.t. the climate data, but some of the methodological choices are still not ideal in my opinion (primarily w.r.t. gridding).

Major Comments:

My concerns about the source of the climate data have been largely addressed with additional explanation and references in the Methods section, although some mention of the fact that most of the humidity data are estimated from minimum temperature data in an earlier section would benefit the reader, I think.

My concern about the resolution and gridding of the climate data was mostly addressed and I appreciate that the authors did provide results based on direct interpolation of climate data to sites, but I do not agree with the author's rationale for gridding.

Based on the data provided, the choice of interpolation does not have a large impact, but I would have liked to see, for example, how many of the GAMM models converged with the site-based interpolation (I suspect maybe more). It would have been good to see t-value maps like those in Fig. 3 with site-based estimates as well.

There is some change apparent in Fig. R3 and I would consider these results closer to the truth than what is shown in Fig. 4 in the manuscript, but I would agree that for the most part, the differences are probably not relevant/meaningful. It is interesting, though, that Mean Age notably decreased in importance (as measured by MSE).

More broadly, however, I strongly disagree with the authors' rationale for choosing gridding in the first place, and I believe that using station-based interpolation would be the methodologically more appropriate approach (and that is what I meant to suggest, not interpolation to a very dense, high-resolution grid, although I guess I could have been clearer). It is true that weather stations in Canada are sparse, but if one is interested in specific sites, there is no point in interpolating to a regular grid first and then choosing the closest point, when one could just use, e.g. IDW to average available stations in the vicinity. This is even more so the case when lapse-rate adjustment is applied. Every additional step of interpolation introduces more uncertainty.

Interpolation to a regular and appropriately-spaced grid is only necessary, when area averages are being considered, where representativeness of a station for an area is of concern.

Concerning the correlation between climate variables on the regular grid and at site locations: I am almost surprised that it is not higher. Correlation only measures variability and does not capture constant biases, but the main difference I would expect between site values and regular grid points would be in the lapse-rate adjustment, which primarily manifests in relatively constant deviations. RMSE or bias would have been a better measure here.

(I am assuming the authors are referring to temporal correlation.)

Similarly

Minor Comments:

Fig. R2: I am not sure what the authors are trying to show... there are clearly areas in the southern part of the domain where station density is much higher than 60 - 100 km and a finer

grid would be justified (Fig. R1). At the same time Fig. R2 also clearly shows that there are regions in the Rocky Mountains where the coarser grid misses regionally differing trends that are present at higher resolution, and which would reduce noise in the analysis, if properly captured.

If the authors are concerned about some regions having lower levels of precision (more noise) than others due to station density: that is already the case due to strong topographic gradients in the Rocky Mountains, and gridding to a 1 degree grid exacerbates this issue.

Figure 1: The figure looks significantly better, but the resolution could still be improved.

Figures 2 & 3: These figures are much improved and I think the decision to move most of the species-dependent plots into SI was good.

What was the selection criterion for the species shown in the main manuscript?

Figure 6: I am not sure if the change in color map does anything to improve the readability of the map. In particular, the large water bodies on the map are shown in a color that is also part of the color map, and at first glance it was not apparent to me that this was simply a geographic overlay - I think a color that is not part of the color map should be used here, or simply a thin black outline. Furthermore, the authors state that the resolution had been increased, however, I struggle to see that.

I think it would also be appropriate to mention in the context of Fig. 6 that the underlying humidity data were largely estimated from Tmin (using Kimball's method).

Reviewer #4 (Remarks to the Author):

As requested by the editor I do not give an additional full review of the paper - I feel the three reviewers before did already an excellent job and identified a whole set of unclarities. A fully agree with my co-referees that the paper provides novel and important information showing that not only in rather warm and already (from the view of the atmosphere and the soil) dry regions (as e.g. by Williams et al. 2013) increases in VPD are already today (and even more in future) a threat for tree growth and functioning. The amount of tree ring data is impressive and it is central that not only species difference but also effects of age and region were accounted for.

The response of the authors to the points raised by the three referees:

As mentioned below the referees not really identified errors in interpretation or flaws in methodology but mainly ask for clarification of several points were either some information was missing in the original manuscript or was described in a rather ambiguous way. In the revised version the author did an excellent job and clarified - in my opinion - all points that were put forward by the reviewers. This holds true for the technical questions of Paolo Cherubini, related to BAI calculations, cross dating and additional inclusion of ITRDB data as well as for the clarification of the VPD and humidity data as well as of the BioSIM estimates as requested by referee 3. From my point of view the results are novel and show the VPD dependency has been shown based on a really large data set and the authors reacted very adequately to the points raised by the reviewer.

Increasing atmospheric dryness reduces boreal forest tree growth

Response to reviewers

Reviewer #3 (Remarks to the Author):

Second Review of Mirabel et al. "Increasing atmospheric dryness reduces boreal forest tree growth"

I appreciate that the authors have made an effort to alleviate my concerns. The required additional information regarding the climate data has now been added to the Methods section and (some of) the figures have been improved.

Based on the information provided, I think it is very likely that the results are robust w.r.t. the climate data, but some of the methodological choice are still not ideal in my opinion (primarily w.r.t. gridding).

We appreciate your time and advice regarding the information added after the first round of reviews. Below, we will address your concerns and provide all the necessary materials to properly support our methodological choices.

Major Comments:

My concerns about the source of the climate data have been largely addressed with additional explanation and references in the Methods section, although some mention of the fact that most of the humidity data are estimated from minimum temperature data in an earlier section would benefit the reader, I think.

We are pleased to hear that the changes we made to the methods have enhanced readability. In accordance with your advice to clarify our VPD calculation method as early as possible in the manuscript, we have applied the suggested changes to the caption of Figure 6 (please see the later comment for details).

My concern about the resolution and gridding of the climate data was mostly addressed and I appreciate that the authors did provide results based on direct interpolation of climate data to sites, but I do not agree with the author's rationale for gridding.

Based on the data provided, the choice of interpolation does not have a large impact, but I would have liked to see, for example, how many of the GAMM models converged with the site-based interpolation (I suspect maybe more). It would have been good to see t-value maps like those in Fig. 3 with site-based estimates as well.

To further justify our choices, we tested the convergence rate of models using the site-based interpolation and corresponding maps. With this site-based dataset, 72% (3,572 models) of the

species-by-site models converged, which is 13 more sites compared to our previous results where we had 3,559 converging models (same percentage).

We also reproduced the maps as was done in Figure 3, following the same procedure. The two maps are nearly indistinguishable from those included in the manuscript.

Fig. R1. Pointwise t-values of the regression between annual growth fluctuations estimated from tree rings (BAI) and vapour pressure deficit (VPD) using a site-based interpolation. Maps are displaying site-species t-values; a bidimensional interpolation was performed on a spatial resolution of 1 x 1 degree, using the inverse distance weighting method based on the 12 closest neighbours. Interpolations were bounded using boreal mask (all species map)

As suggested, we proceeded with a double-checking process and obtained a similar number of convergent sites and similar t-value maps. It can be concluded that the use of climate data extracted at the site scale has no different effect compared to the use of climate data extracted at the nearest grid point for the specific statistical models used in our study.

There is some change apparent in Fig. R3 and I would consider these results closer to the truth than what is shown in Fig. 4 in the manuscript, but I would agree that for the most part, the differences are probably not relevant/meaningful. It is interesting, though, that Mean Age notably decreased in importance (as measured by MSE).

It is true that the validation dataset yielded slightly modified results. However, these modifications did not lead us to alter our conclusions, as the relative importance of each variable remained fundamentally unchanged. Since site-specific data were utilized in the validation process, slight variations in MAT and MAP values could affect the random forest modeling. Pending the editor's approval, we would prefer to maintain the gridded approach, given the robustness of the results.

More broadly, however, I strongly disagree with the authors rational for choosing gridding in the

first place, and I believe that using station-based interpolation would be the methodologically more appropriate approach (and that is what I meant to suggest, not interpolation to a very dense, hi-res grid, although I guess I could have been clearer). It is true that weather stations in Canada are sparse, but if one is interested in specific sites, there is no point in interpolating to a regular grid first and then choosing the closest point, when one could just use, e.g. IDW to average available stations in the vicinity. This is even more so the case when lapse-rate adjustment is applied. Every additional step of interpolation introduces more uncertainty. Interpolation to a regular and appropriately-spaced grid is only necessary, when area averages are being considered, where representativeness of a station for an area is of concern.

The use of gridded data in dendroclimatological studies is a standard practice. This format makes it easier to distribute and share the data with other researchers, as it is structured and standardized. Gridded data often results in a smaller dataset compared to using raw, point-level data. This reduction in data size can lead to more efficient computing. As we have illustrated here, this choice has no impact on the results. In the context of the site-level modeling approach described in this work, the choice of gridded data does not introduce any significant bias or affect the outcomes. We acknowledge that the impact of using gridded data may vary depending on the modeling approach. For instance, if a different modeling approach, such as using Generalized Additive Mixed Models (GAMM), were applied to range-wide species data, the results might be slightly noisier (less precise or more variable) when using gridded data.

Concerning the correlation between climate variables on the regular grid and at site locations: I am almost surprised that it is not higher. Correlation only measures variability and does not capture constant biases, but the main difference I would expect between site values and regular grid points would be in the lapse-rate adjustment, which primarily manifest in relatively constant deviations. RMSE or bias would have been a better measure here. (I am assuming the authors are referring to temporal correlation.)

See responses to earlier comments in regard to the use of gridded data. Yes, we were referring to temporal correlation.

Minor Comments:

Fig. R2: I am not sure what the authors are trying to show... there are clearly areas in the southern part of the domain where station density is much higher than 60 - 100 km and a finer grid would be justified (Fig. R1). At the same time Fig. R2 also clearly shows that there are regions in the Rocky Mountains where the coarser grid misses regionally differing trends that are present at higher resolution, and which would reduce noise in the analysis, if properly captured.

There is seemingly greater density, but the number of stations that cover the full period is low. Note too that we don't have extensive tree-ring data in those areas of higher station density. Hence, there represent little weight in our analyses.

If the authors are concerned about some regions having lower levels of precision (more noise) than others due to station density: that is already the case due to strong topographic gradients in the Rocky Mountains, and gridding to a 1 degree grid exacerbates this issue.

See responses to earlier comments in regard to the use of gridded data.

Figure 1: The figure looks significantly better, but the resolution could still be improved.

We appreciate your comment and have attempted another method to export the maps. We hope the current version is improved. If it is the PDF saving process that is causing file compression, we can provide separate files as per the editor's suggestion.

Figures 2 & 3: These figures are much improved and I think the decision to move most of the species-dependent plots into SI was good.

What was the selection criterion for the species shown in the main manuscript?

The species featured in the main manuscript hold significant economic importance as they are among the primary species harvested. Additionally, their geographic distribution is particularly threatened by climatic changes. For instance, black spruce, which grows in permafrost soils, is expected to be affected by warming and may experience more frequent insect outbreaks. (Gauthier, et al. 2014)

We mentioned these criterion in the methods section:

l. 424: *“Annual fluctuations in growth over time were computed for Picea mariana, Picea glauca, Pinus contorta and Picea engelmannii that are among the primary harvested species in Canada, and which geographic distribution is particularly threatened by climatic changes (Gauthier et al., 2014). Annual growth fluctuations were obtained from the detrending of BAI series that removed growth trends due to tree age and size (Zhang et al. 2018).”*

Figure 6: I am not sure if the change in color map does anything to improve the readability of the map. In particular, the large water bodies on the map are shown in a color that is also part of the color map, and at first glance it was not apparent to me that this was simply a geographic overlay - I think a color that is not part of the color map should be used here, or simply a thin black outline. Furthermore, the authors state that the resolution had been increased, however, I struggle to see that.

The figure was revised accordingly with the suggestions.

I think it would also be appropriate to mention in the context of Fig. 6 that the underlying humidity data were largely estimated from T_{min} (using Kimball's method).

Following your advice, we added the method used for VPD calculation in the figure legend :

l. 704: *« Temporal trends in summer VPD (kPa / year) since 1951, estimated from daily temperature and precipitation using Kimball's method (Kimball et al., 1997).”*

Reviewer #4 (Remarks to the Author):

As requested by the editor I do not give an additional full review of the paper - I feel the three reviewers before did already an excellent job and identified a whole set of unclarities. A fully agree with my co-referees that the paper provides novel and important information showing that not only in rather warm and already (from the view of the atmosphere and the soil) dry regions (as e.g. by Williams et al. 203) increases in VPD are already today (and even more in

future) a threat for tree growth and functioning. The amount of tree ring data is impressive and it is central that not only species difference but also effects of age and region were accounted for.

The response of the authors to the points raised by the three referees:

As mentioned below the referees not really identified errors in interpretation or flaws in methodology but mainly ask for clarification of several points were either some information was missing in the original manuscript or was described in a rather ambiguous way. In the revised version the author did an excellent job and clarified - in my opinion - all points that were put forward by the reviewers. This holds true for the technical questions of Paolo Cherubini, related to BAI calculations, cross dating and additional inclusion of ITRDB data as well as for the clarification of the VPD and humidity data as well as of the BioSIM estimates as requested by referee 3. From my point of view the results are novel and show the VPD dependency has been shown based on a really large data set and the authors reacted very adequately to the points raised by the reviewer.

We appreciate your thorough review of our manuscript and your response to the previous round of reviews. We would also like to express our gratitude for placing our work in perspective and highlighting its relevance.